∂ | **Open Peer Review** | Epidemiology | Research Article

# Spatial patterns of *Hyalomma marginatum*-borne pathogens in the Occitanie region (France), a focus on the intriguing dynamics of *Rickettsia aeschlimannii*

Charlotte Joly-Kukla,[1,2] Célia Bernard,[1,3,4] David Bru,[1] Clémence Galon,[2] Carla Giupponi,[1,3] Karine Huber,[1] Hélène Jourdan-Pineau,[1,3] Laurence Malandrin,[5] Ignace Rakotoarivony,[1,3] Camille Riggi,[1] Laurence Vial,[1,3] Sara Moutailler,[2] Thomas Pollet[1]

**ABSTRACT**  *Hyalomma marginatum* is an invasive tick species recently established in mainland southern France. This tick is known to host a diverse range of human and animal pathogens. While information about the dynamics of these pathogens is crucial to assess disease risk and develop effective monitoring strategies, few data on the spatial dynamics of these pathogens are currently available. We collected ticks in 27 sites in the Occitanie region to characterize spatial patterns of *H. marginatum*-borne pathogens. Several pathogens have been detected: *Theileria equi* (9.2%), *Theileria orientalis* (0.2%), *Anaplasma phagocytophilum* (1.6%), *Anaplasma marginale* (0.8%), and *Rickettsia aeschlimannii* (87.3%). Interestingly, we found a spatial clustered distribution for the pathogen *R. aeschlimannii* between two geographically isolated areas with infection rates and bacterial loads significantly lower in Hérault/Gard departments (infection rate 78.6% in average) compared to Aude/Pyrénées-Orientales departments (infection rate 92.3% in average). At a smaller scale, *R. aeschlimannii* infection rates varied from one site to another, ranging from 29% to 100%. Overall, such high infection rates (87.3% on average) and the effective maternal transmission of *R. aeschlimannii* might suggest a role as a tick symbiont in *H. marginatum*. Further studies are thus needed to understand both the status and the role of *R. aeschlimannii* in *H. marginatum* ticks.

**IMPORTANCE**  Ticks are obligatory hematophagous arthropods that transmit pathogens of medical and veterinary importance. Pathogen infections cause serious health issues in humans and considerable economic loss in domestic animals. Information about the presence of pathogens in ticks and their dynamics is crucial to assess disease risk for public and animal health. Analyzing tick-borne pathogens in ticks collected in 27 sites in the Occitanie region, our results highlight clear spatial patterns in the *Hyalomma marginatum*-borne pathogen distribution and strengthen the postulate that it is essential to develop effective monitoring strategies and consider the spatial scale to better characterize the circulation of tick-borne pathogens.

**KEYWORDS**  *Hyalomma marginatum*, spatial patterns, *Rickettsia aeschlimannii*, symbionts, tick-borne pathogens

*H*yalomma marginatum is a tick species that has recently become established in mainland southern France (1) although it has been established for decades on the French Mediterranean island of Corsica (2–4). *H. marginatum* is currently endemic in several countries from the Maghreb to the Iberian Peninsula and the eastern Mediterranean basin, including Turkey and the Balkans (5). With the increase in temperatures due to climate change, this tick species may become established in northern latitudes via animal movements and/or bird migrations (1, 6, 7). It appears that *H. marginatum* is

Address correspondence to Charlotte Joly-Kukla, charlotte.joly@cirad.fr, Sara Moutailler, sara.moutailler@anses.fr, or Thomas Pollet, thomas.pollet@inrae.fr.

The authors declare no conflict of interest.

present in regions with specific climate features such as warm temperatures in summer and low precipitation which are typical of the Mediterranean climate (8, 9). In mainland France, *H. marginatum* establishment has been reported in several departments including Pyrénées-Orientales, Aude, Hérault, Gard, Var, Ardèche, and Drôme (8). Since an invasion process seems to be happening in the south of France, its adaptation abilities and its current expansion area in France are being closely monitored.

Like many other tick species, *H. marginatum* harbors complex microbial communities, collectively known as the microbiota encompassing symbionts, commensals, environmental microbes, and on the other hand, pathogens affecting vertebrate hosts. Among the pathogens, *H. marginatum* can carry a large diversity of bacteria, viruses, and parasites (10). In Afrotropical regions, the Mediterranean basin, and in France, *H. marginatum* is considered the main candidate for the transmission of the deadly Crimean-Congo hemorrhagic fever virus (CCHFV) (11, 12). CCHFV was detected for the first time in France in ticks from this study and in ticks collected in 2023 (13).

The risk represented by this tick species requires an exhaustive identification of pathogens it carries and the characterization of their dynamics in both space and time. Based on previous studies, *H. marginatum* distribution in the Occitanie region (NUTS-1) seems to be clustered between two geographic areas, in Gard/ northern Hérault departments, and in Pyrénées-Orientales/south Aude departments (8). No *H. marginatum* was found so far between these two geographic clusters for unclear reasons. In this context, we hypothesized that the spatial distribution of microbial communities might vary depending on the geographic cluster. It is now accepted that spatial patterns at different scales impact tick distribution, density, and their associated pathobiome, due to environmental characteristics such as abiotic factors (temperature, landscape, vegetation) and the presence of hosts in given areas forming specific environmental niches (14, 15). The small spatial scale is important to consider as the pathogen prevalence can vary among a given biotope, probably due to specific environmental factors (16). At a larger spatial scale, the type of habitat can also participate in the prevalence of pathogens as was the case for *Anaplasma phagocytophilum* and *Rickettsia* sp. of the spotted fever group infections in *Ixodes ricinus* collected in Central France from two different habitats (pasture vs woodland) (17). Another study reported differences in *Borrelia burgdorferi* prevalences between urban/suburban habitats and natural/agricultural in Slovakia (18).

While a previous study identified *H. marginatum*-borne pathogens in several sites of collection in the South of France between 2016 and 2019 (19), information about the dynamics of these pathogens remains particularly scarce. We thus propose in this study a focus on the spatial distribution of *H. marginatum*-borne pathogens in Occitanie, by taking into account the influence of other factors such as the tick sex and the engorgement status (20–23). To this purpose, a large-scale tick collection program was performed in the Occitanie region in May 2022 and resulted in the analysis of 510 ticks found in 27 sites across four departments. We characterized the influence of spatial patterns, tick sex, and engorgement status on tick-borne pathogens' infection rates and loads.

## RESULTS

### Genotyping of microbes detected in *H. marginatum*

On the 510 ticks analyzed using the BioMark assay, 445 samples were positive for *Rickettsia aeschlimannii*. On the six sequences that were obtained, the blast analysis of the longest sequence (GenBank Accession Number: PP236764) showed 100% identity with a sequence of *R. aeschlimannii* obtained from a *H. marginatum* tick in England in 2019 (AN: MT365092.1) and 100% identity with another sequence obtained from a tick collected on a horse in 2019 in the Gard department of the Occitanie region (19) (AN: PP379722). The results allowed the identification of *R. aeschlimannii* and it was extrapolated to the other samples exhibiting identical amplification patterns. Two species of *Anaplasma* were detected: eight samples for *A. phagocyotophilum* and four for *A. marginale*. The three sequences from samples positive for *A. phagocytophilum* corresponded to *A. phagocytophilum*; the longest sequence (AN: PP265050) showed

100% identity with a sequence isolated in 2008 from *I. ricinus* in Italy (JQ669948.1). Finally, all four sequences of positive samples for *A. marginale* allowed the identification of *A. marginale*, with the longest sequence (AN: PP218690) showing 99.4% identity with a sequence obtained from infected cattle in Iran (GenBank AN: MK016525.1). *Theileria equi* was detected in nine samples. One sequence was obtained (AN: PP227163) and resulted in 100% identity with a sequence of *T. equi* obtained from an infected *Rhipicephalus bursa* (MK732476.1) in Corsica, France. *Theileria orientalis* was detected in one tick (AN: PP358744), and the sequence showed 99.7% identity with a sequence from a *Haemaphysalis longicornis* tick in China in 2014 (MH208633.1).

## Infection rate of *H. marginatum* microbes

Among the 510 ticks analyzed with the BioMark assay and quantitative PCR (qPCR), 11.8% (95% CI: 9.0%–14.6%) were infected with at least one pathogen except *R. aeschlimannii* (*A. phagocytophilum, A. marginale, T. equi,* or *T. orientalis*). Infection rates using the BioMark assay for *A. phagocytophilum* and *A. marginale* were, respectively, 1.6% (0.5%–2.7%) and 0.8% (0.02%–1.6%). *T. equi* and *T. orientalis* infection rates were, respectively, 1.8% (0.6%–2.9%) and 0.6% (0%–0.6%). Finally, *R. aeschlimannii* and *Francisella*-LE infection rates reached, respectively, 87.3% (84.4%–90.2%) and 96.7% (95.1%–98.2%). Re-evaluated *R. aeschlimannii* and *T. equi* infection rates by qPCR were, respectively, 89.4% (86.7%–92.1%) and 9.2% (6.7%–11.7%). *R. aeschlimannii* infection rate was similar with respect to the detection method used whereas the re-evaluation of *T. equi* infection rate using the qPCR was about five times higher. One co-infection in one tick was reported between *A. marginale* and *T. orientalis,* which represented an infection rate of 0.2% (0%–0.6%). *Francisella*-LE and *R. aeschlimannii* were not considered in co-infections.

## Multivariate analyses of *H. marginatum* microbes

### *Spatial patterns*

#### *According to the geographic clusters*

The tick-borne pathogen dynamics were first assessed by defining a spatial structure linked to the geographic distribution of ticks which consists of two separate areas in the departments Hérault/Gard (HG) and Aude/Pyrénées-Orientales (APO) (Fig. 1). *R. aeschlimannii* infection rate was significantly influenced by the geographic cluster ($\chi^2$ = 4.1210; df = 1; *P*-value = 0.04235, Fig. 2A), it was higher in the cluster APO: 92.3% (89.3%–95.2%) compared to the cluster HG: 78.6% (72.7%–84.5%). *T. equi*, *Francisella*-LE, and *A. phagocytophilum* infection rates were not significantly influenced by the geographic cluster (Table S2). The influence of the geographic cluster on *R. aeschlimannii* and *T. equi* loads was also analyzed. *R. aeschlimannii* bacterial loads were significantly influenced by the geographic cluster ($\chi^2$ = 6.5292; df = 1; *P*-value = 0.01061, Fig. 2B). Indeed, bacterial loads were significantly higher in APO: $7.6 \times 10^5$ genome.$\mu$L$^{-1}$ in average ($6.5 \times 10^5$ – $8.6 \times 10^5$ genome.$\mu$L$^{-1}$) compared to HG: $2.0 \times 10^5$ genome.$\mu$L$^{-1}$ in average ($1.4 \times 10^5$ – $2.6 \times 10^5$ genome.$\mu$L$^{-1}$). While *T. equi* infection rates were not influenced by the geographic cluster, this spatial variable significantly influenced the *T. equi* loads ($\chi^2$ = 5.4868; df = 1; *P*-value = 0.01916) (Table S3) as the parasite loads were significantly higher in APO: $1.0 \times 10^3$ genome/$\mu$L in average ($0 – 3.1 \times 10^3$ genome.$\mu$L$^{-1}$) compared to HG: 5.3 genome.$\mu$L$^{-1}$ in average (2.9 – 7.7 genome.$\mu$L$^{-1}$).

#### *According to the collection sites*

The collection site had a significant influence on *R. aeschlimannii* infection rate (*P*-value = $3.88 \times 10^{-29}$) as illustrated by a variability ranging from 29% to 100% between one site to another in Occitanie. In the cluster HG, 50% of the sites had an infection rate up to 100% compared to 84% of the sites in the cluster APO. Interestingly, infection rates for sites located in the geographic cluster HG were more variable than those from the cluster

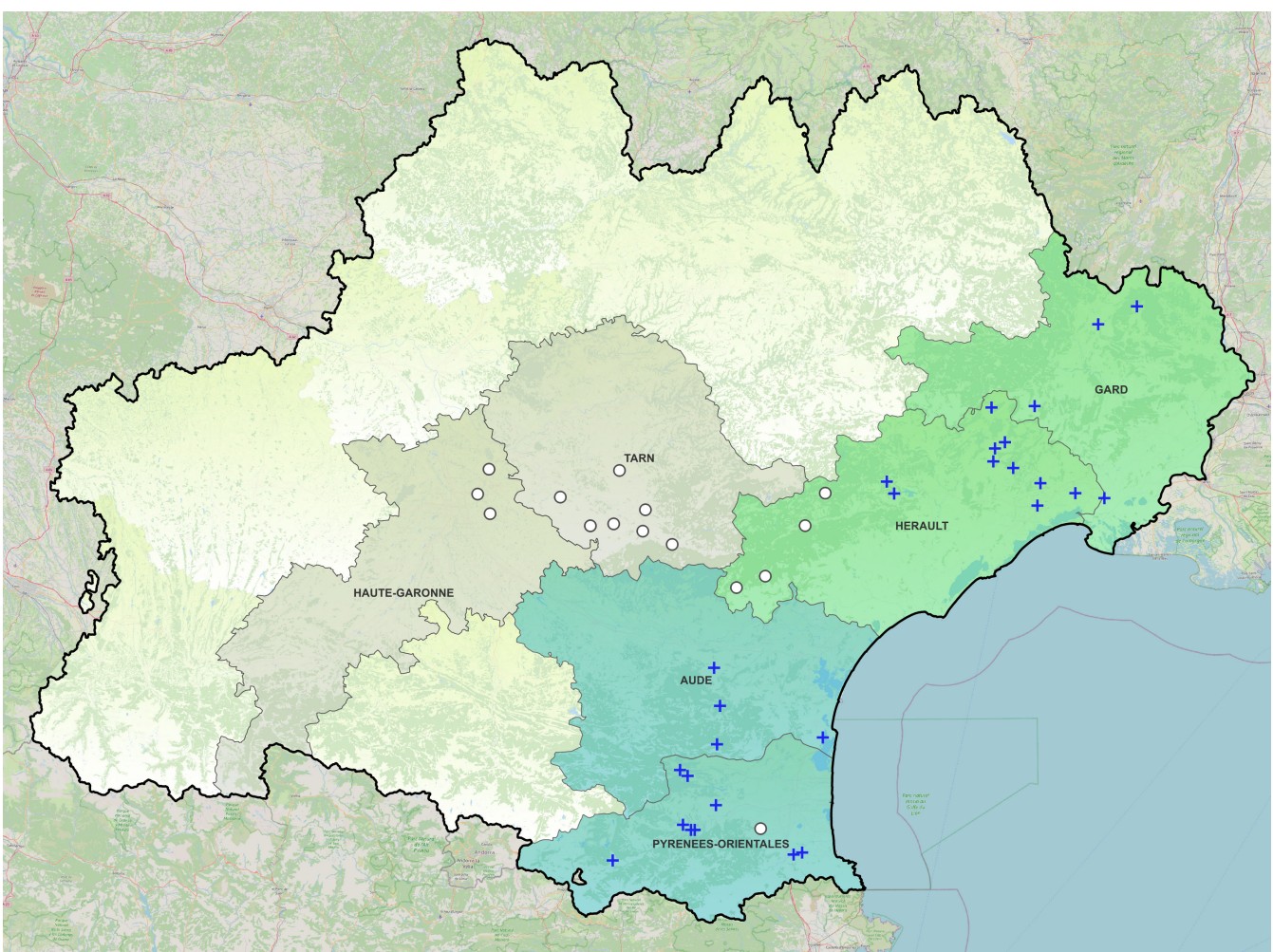

**FIG 1** Map of the region Occitanie where the tick sampling was performed in May 2022. The Occitanie region is delimited by a black line. White circles indicate *H. marginatum*-free sites and blue crosses indicate *H. marginatum*-positive sites where ticks were collected and selected for analyses in 2022. Blue-colored departments represent the geographic cluster Aude/Pyrénées-Orientales and the green-colored departments represent the geographic cluster Hérault/Gard. Gray-colored departments were visited but no *H. marginatum* ticks were found. Uncolored areas correspond to other departments that were not visited for the tick collection since *H. marginatum* introduction/installation was not reported (5). The map was created using QGIS.org, 2024. QGIS Geographic Information System. QGIS Association. http://www.qgis.org.

APO (Fig. 3A and B). The collection site did significantly influence *T. equi* infection rate ($P$-value = $3.05 \times 10^{-8}$), ranging from 0% to 41% from one site to another in Occitanie (Fig. 4A). Finally, it did not have a significant influence on *Francisella*-LE ($P$-value = 0.707) and *A. phagocytophilum* ($P$-value = 0.055) infection rates although they varied, respectively, from 80% to 100% and 3.3% to 20% (Fig. 4B and C). Finally, *A. marginale* infected rates were quite similar between the two sites of collection where it was detected (3.1% and 8.8%) (Fig. 4D).

## The sex of the ticks

While infection rates estimated for *R. aeschlimannii*, *T. equi,* and *A. phagocytophilum* were not significantly influenced by the sex of the ticks (Table S2), slight but significant differences were observed for *Francisella*-LE between males and females ($x^2$ = 4.4239; df = 1; $P$-value = 0.03544) as *Francisella*-LE infection rate was significantly higher in females: 98.7% (97.3%–100%) compared to males 94.9% (95% CI: 92.4%–97.5%). Finally, even though the sex of ticks did not influence the *R. aeschlimannii* infection rates (Fig. 2C), significant differences were observed for *R. aeschlimannii* loads ($x^2$ = 15.6199; df =

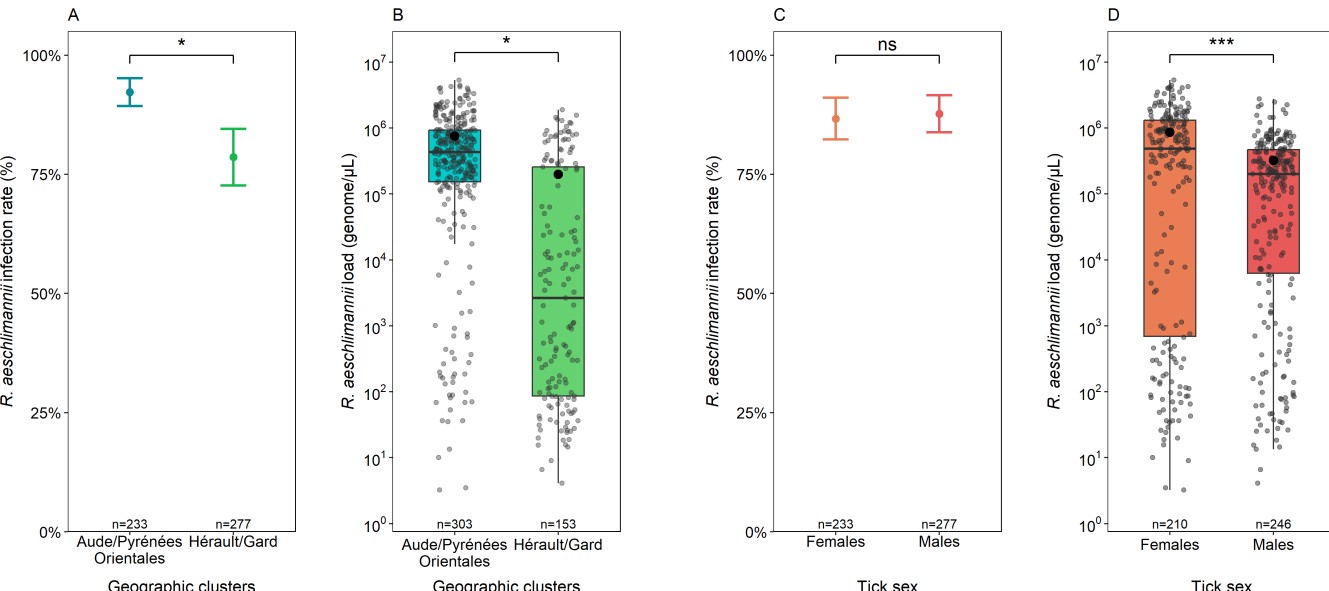

**FIG 2** Infection rate and bacterial load of *R. aeschlimannii* in *H. marginatum* ticks collected in the two geographic clusters Hérault/Gard and Aude/Pyrénées-Orientales (A and B) and according to the tick sex (C and D). The number of ticks per cluster is indicated above the x-axis legends. Significant differences are represented by asterisks (*, *P*-value < 0.05). (A and C) *R. aeschlimannii* presence (1) or absence (0) for each tick examined by the BioMark assay is summarized by the mean (infection rate in %) and error bars represent a 95% confidence interval. (B and D) Bacterial loads expressed in genome.µL$^{-1}$ obtained by qPCR are represented by a boxplot summarizing the median, first, and third quartiles. Each gray dot represents the load of *R. aeschlimannii* for one tick. The mean is symbolized by the black dot.

1; *P*-value = 7.447 × 10$^{-5}$ (Fig. 2D; Table S3). Bacterial loads were significantly higher in females: 8.6 × 10$^5$ genome.µL$^{-1}$ in average (7.1 × 10$^5$ – 1.0 × 10$^6$ genome.µL$^{-1}$) compared to males: 3.2 × 10$^5$ genome.µL$^{-1}$ in average (CI: 95% 2.7 × 10$^5$ – 3.8 × 10$^5$ genome.µL$^{-1}$). By contrast, *T. equi* loads were not significantly impacted by the sex of the ticks (Table S3).

## The engorgement status

The infection rate of *T. equi* in female ticks was significantly influenced by the engorgement status ($\chi^2$ = 8.8679; df = 2; *P*-value = 0.01187). It was higher in fed females compared to the unfed ones with values reaching 1.8% on average (0%–4.4%) for unfed females and 14.9% on average (6.6%–23.2%) for fed females. Finally, *Francisella*-LE and *R. aeschlimannii* infection rates as well as *R. aeschlimannii* and *T. equi* loads were not significantly influenced by the engorgement status (Tables S2 and S3). No statistical analysis of the impact of the engorgement status on *A. phagocytophilum* could be performed due to a low number of positive samples for this pathogen (Table S2).

## The hosts

No influence of the tick host was observed on the infection rates of *R. aeschlimannii* ($\chi^2$ = 0.028; df = 1; *P*-value = 0.86715), nor *T. equi* ($\chi^2$ = 0.0184; df = 1; *P*-value = 0.892123). The host did show a significant influence on *A. phagocytophilum* infection rate ($\chi^2$ = 5.6442; df = 1; *P*-value = 0.01751). *A. phagocytophilum* infection rate was higher in cattle compared to horses. In APO, 6/7 positive ticks were collected on cattle and 1/7 on horses. No statistical analysis was conducted for *A. marginale* because of too low positive samples but 4/4 *A. marginale* positive ticks were collected on cattle.

## Maternal transmission of *R. aeschlimannii*

*R. aeschlimannii* was detected in all five egg-laying *H. marginatum* females with bacterial loads estimated to be 2.0 × 10$^6$ genome.µL$^{-1}$ on average (0 – 5.7 × 10$^6$ genome.µL$^{-1}$). Hundred percent of the egg pools, belonging to each egg-laying female, were positive

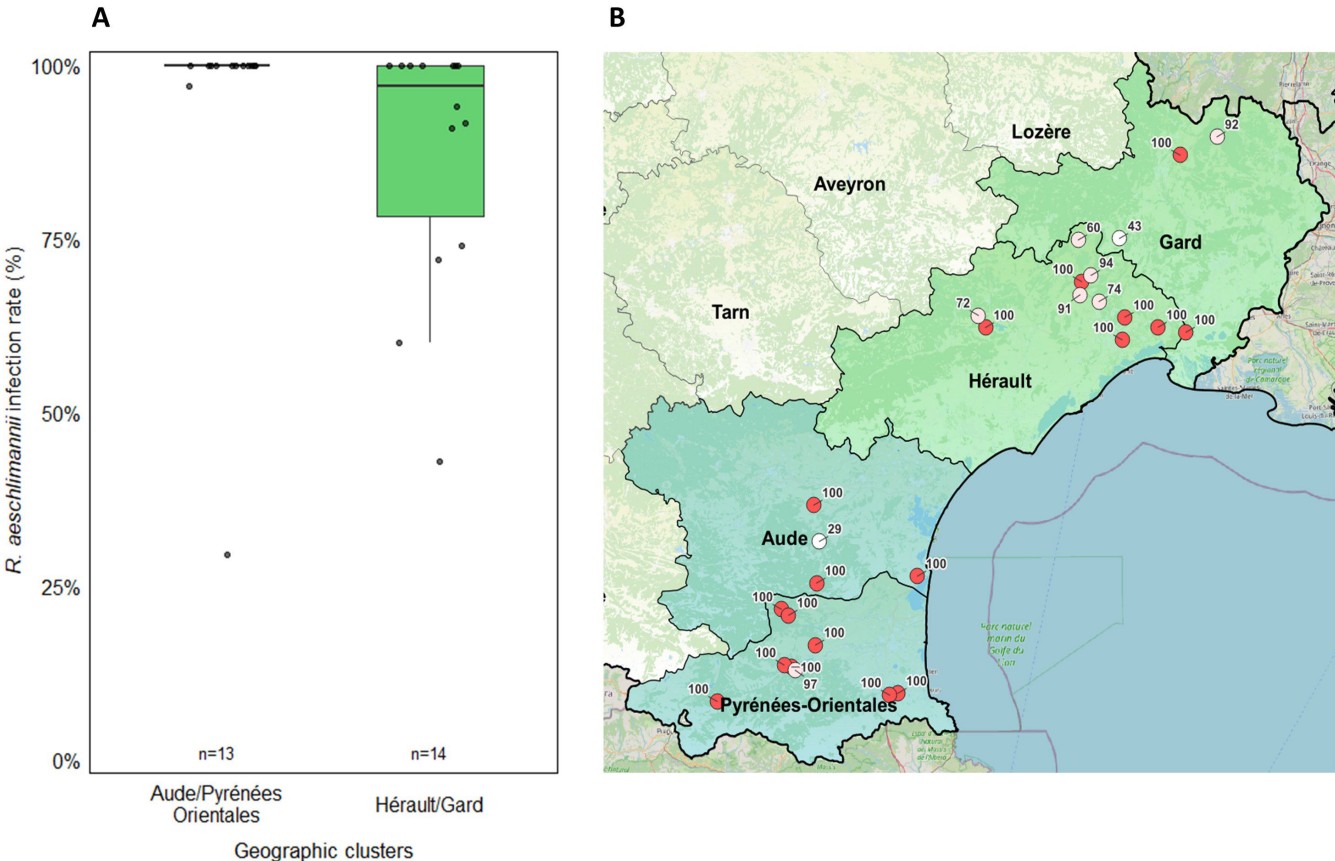

**FIG 3** (A) *R. aeschlimannii* Infection rate in *H. marginatum* for each site of collection in the two geographic clusters Hérault/Gard and Aude/Pyrénées-Orientales. The number of sites is indicated above the x-axis legends. Infection rates for each site were determined with the BioMark assay and are represented by black dots. (B) *R. aeschlimannii* spatial distribution and its infection rates for each site; represented by a red circle when the infection is 100% and in white when it is less than 100%. The mean infection rates per site is indicated next to each site.

for *R. aeschlimannii* with bacterial loads estimated to be $3.3 \times 10^5$ genome.µL$^{-1}$ in average ($2.2 \times 10^5 - 4.5 \times 10^5$ genome.µL$^{-1}$). The only one pool of 12 larvae was also positive for *R. aeschlimannii* with bacterial load estimated to be $1.1 \times 10^5$ genome.µL$^{-1}$.

## DISCUSSION

Recently established in the south of France and known to potentially transmit human and animal pathogens, *H. marginatum* might represent a future problem for both public and animal health. A better assessment of the risk linked to this tick first requires an exhaustive identification of *H. marginatum*-borne pathogens and their dynamics. In this context, the objective of this study was to characterize the influence of spatial patterns on both the infection rates and loads of *H. marginatum*-borne pathogens in the Occitanie region in France.

In our study, we detected *R. aeschlimannii, T. equi, T. orientalis, A. phagocytophilum,* and *A. marginale* in adult *H. marginatum* ticks. The global infection rates of these pathogens across the Occitanie region were 87.3%, 9.2%, 0.2%, 1.6%, and 0.8%, respectively. Most of the detected pathogens corresponded to microbes known to have circulated in this geographic area over the last 5 years (19) which would suggest a certain stability in the circulation of the main *H. marginatum*-borne pathogens in these departments. However, we did not detect the bacteria *Ehrlichia minasensis* previously reported at very low prevalence (in a single tick collected from a horse in the Occitanie region) (19). We detected *T. orientalis,* responsible for benign theileriosis in cattle (24), in a fed female collected on a bovine in a single site in Pyrénées-Orientales. This

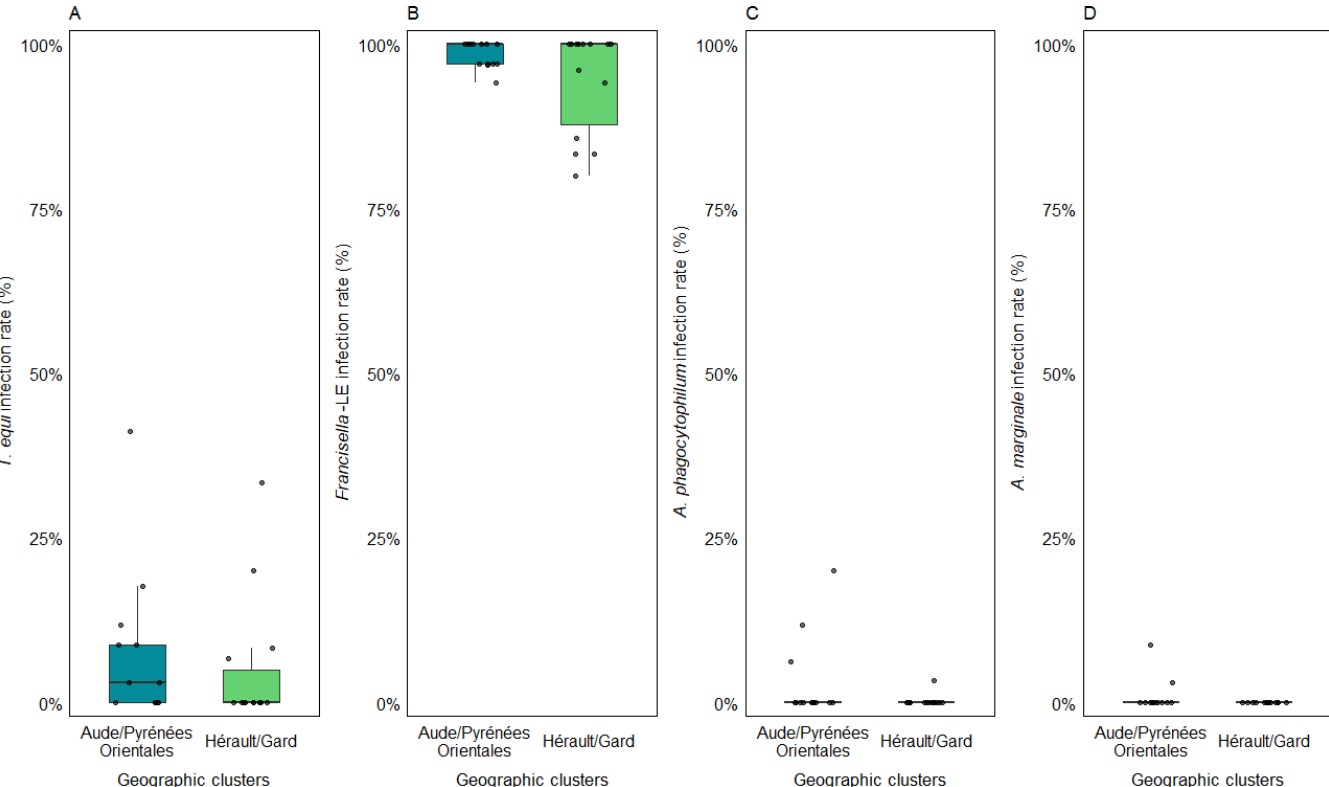

**FIG 4** Infection rates of *T. equi* (A), *Francisella*-LE (B), *A. phagocytophilum* (C), and *A. marginale* (D) in *H. marginatum* ticks for each site of collection in the two geographic clusters Hérault/Gard (*n* = 14 sites) and Aude/Pyrénées-Orientales (*n* = 13 sites). Infection rates for each site represented by black dots were determined with the BioMark assay for *Francisella*-LE, *A. phagocytophilum,* and *A. marginale* and with the qPCR assay for *T. equi*. The map was created using QGIS.org, 2024. QGIS Geographic Information System. QGIS Association. http://www.qgis.org.

detection was particularly unexpected as this parasite was not previously reported in the area. Furthermore, pathogen infection rates were variable across the sites. In epidemiological terms, these results underline the regular monitoring (e.g., multi-year surveys) of *H. marginatum*-borne pathogens at fine spatial scales in order to detect pathogens circulating insidiously in the studied area. Please refer to Bernard et al. (19) for more details on the vector competence discussion of the detected pathogens in *H. marginatum*.

Excluding *R. aeschlimannii* due to the very high infection rate estimated in ticks for this bacterium (see below for specific discussion on *R. aeschlimannii*), 11.8% of ticks were positive for at least one of the other tested pathogens and only one co-infection was identified between *A. marginale* and *T. orientalis* (0.2%). This low number of co-infections contrasts with *I. ricinus* which has up to five different co-infections (25). Co-infections are known to enhance disease severity as is the case for babesiosis and Lyme disease (26). Overall, while *H. marginatum* is known to carry a wide range of pathogens (10), we detected a low diversity of pathogens circulating in ticks in the Occitanie region with two *Anaplasma* species, two *Theileria* species, one species of *Rickettsia* and only one co-infection. *H. marginatum* carries fewer pathogens than other tick species of public and veterinary importance like *I. ricinus,* due to differences in their life cycle such as the number and the host spectrum (27, 28). This substantial number of positive ticks for pathogens known to potentially affect both human and animal health does incite to accentuate the prevention and maintenance of regular surveillance of *H. marginatum*-borne pathogens in this area.

## The scheming dynamics of *R. aeschlimannii*

*R. aeschlimannii* is known as an agent of spotted fever rickettsiosis, a human infection that mainly occurs in North and South Africa, Greece, Italy, and Germany. This bacterium was detected in 1997 in Morocco for the first time in *H. marginatum* (29), which is considered to be the reservoir and a potential vector although no vector competence experiment was conducted (10, 30). Buysse et al. showed that 324 genes from the genome of *R. aeschlimannii* of *H. marginatum* ticks collected from Italy cluster with the human pathogen strain, indicating that this bacterium is pathogenic for humans and potentially symbiotic for *H. marginatum (*31*)*. Further studies are required to characterize the full genome of *R. aeschlimannii* in *H. marginatum* ticks collected from France in order to compare it with the genome of pathogenic *R. aeschlimannii*. This will help to determine whether the bacterium encountered in ticks is the one responsible for human rickettsiosis cases or whether it is a different strain, potentially involved in a symbiotic relationship with the tick. In our study, *R. aeschlimannii* infection rates were very high whatever the detection method and the targeted gene, the citrate synthase gene through the BioMark assay (87.3%) or with the *ompB* gene through qPCR (89.4%). These results are consistent with previous studies performed on *H. marginatum* in southern France which reported high infection rates; from 75% of individual ticks in *H. marginatum* collected around the French Mediterranean Sea between 2016 and 2019 (19) to 100% of the pools of *H. marginatum* in Corsica (4). While high *R. aeschlimannii* infection rates are observed in our study and many others (4, 19, 32, 33), a low number of human rickettsiosis cases are observed (34–36), which questions the vector capacity of *H. marginatum* for this bacterium. The presence of *R. aeschlimannii* in the *H. marginatum* salivary glands would be informative to know if it is a strain that could be transmitted to the tick host and may be responsible for human cases of rickettsiosis.

Several of our results raise the question of whether *R. aeschlimannii* is a symbiotic bacterium in *H. marginatum*. Indeed, the transovarian transmission of *R. aeschlimannii* in *H. marginatum* paired with its high infection rates both support this assumption (37, 38). In the literature, such hypotheses have already been suggested for *H. marginatum* (32, 39). Tick symbionts are typically classified as primary or secondary symbionts, depending on their role in the tick's survival. Its status as a primary symbiont seems unlikely because of the significant spatial variability in terms of infection rates and loads, where the infection rate of *Francisella*-like symbiont is very stable regardless of spatial patterns. Therefore, *R. aeschlimannii* would rather be a secondary symbiont. Overall, this hypothesis contrasts with the postulate that *Rickettsia* symbionts are most common in ticks of the genera of *Ixodes*, *Amblyomma*, and *Dermacentor*, whereas it has been less frequently found in *Rhipicephalus*, *Haemaphysalis*, and *Hyalomma* ticks (40–42). Regarding the functions of symbiotic *Rickettsia* in tick physiology, it was already demonstrated that they can be involved in nutrition, such as the *Rickettsia buchneri* symbiont in *Ixodes scapularis* or *Ixodes pacificus* (43, 44), as it harbors all required genes for folate biosynthesis (45). This nutritive role is unlikely in the case of *H. marginatum,* since *R. aeschlimannii* do not have complete biosynthesis pathways for B vitamins, as opposed to the two primary symbionts *Francisella*-LE and Midichloria that possess either complete biotin, riboflavin, or folic acid pathways (31). Other functions of *Rickettsia* in arthropods in the literature mention defense (46) and reproductive manipulation (47). Our results indicate similar infection rates whatever the tick sex assuming that the function of *R. aeschlimannii* appears to benefit both male and female ticks (whether fed or not). Its role in reproductive manipulation seems unlikely since the high proportion observed in males. In regard to the present information, we suggest that *R. aeschlimannii* would be involved in tick defense against abiotic factors, and further investigations for example using transcriptomics approaches should be conducted in the future to confirm such a hypothesis.

In the present study, strong spatial patterns were observed for *R. aeschlimannii*, whether it be at a large spatial scale (geographic cluster) or at a small one (from one site to another, especially in Hérault/Gard). Some of the infection rate variability

between the sites might be explained by unequal tick numbers per site. It is also possible that while *R. aeschlimannii* is present in ticks since it is maternally transmitted, the quantities are too low to be detected efficiently, contributing to this variability. In this study, we thus propose two hypotheses to explain this uneven spatial distribution of *R. aeschlimannii* infection rates. The first is to assume that the *R. aeschlimannii* infection rate can be influenced by environmental characteristics (humidity, temperature, vegetation, host variability). *R. aeschlimannii* would replicate preferentially when environmental conditions threaten the tick and this would be in Aude/Pyrénées-Orientales rather than in Hérault/Gard. In the literature, it was reported that infection by *Rickettsia* sp. of the spotted fever group in *I. ricinus* is influenced by the geographic location and their environmental characteristics (forest fragmentation, vegetation, hosts) (17, 48). Another study reported a significant difference in loads of *Rickettsia* phylotype G021 between *I. pacificus* from different collection sites and vegetation habitats (49). In order to explore this hypothesis, it would be interesting to evaluate the infection rate by *R. aeschlimannii* and the loads from ticks experimentally maintained in microcosms located in different environmental conditions (vegetation, temperature) in the field (50), in order to evaluate the influence of these variables and provide an insight into a potential function of this bacterium for the tick. The second hypothesis which is not in conflict with the first, is that the presence and loads of *R. aeschlimannii* would vary from one *H. marginatum* population to another. It is interesting to note that, the spatial distribution of *R. aeschlimannii* in the two geographic areas (Hérault/Gard and Aude/Pyrénées-Orientales) is consistent with two genetically differentiated populations of *H. marginatum* characterized using mitochondrial markers clusters main haplotypes (Giupponi et al., personal communication). The presence of structured genetic differentiation could be explained by the introduction of tick populations by migratory birds or horses and cattle from different geographic origins. These introduced tick populations could harbor microbial communities with different compositions, which could explain the spatial patterns of *R. aeschlimannii* in French ticks.

## Dynamics of other *H. marginatum* microbes

As expected, *Francisella*-LE was detected in almost all ticks (96.7%). *Francisella*-LE is a known primary symbiont that produces B vitamins vital for tick survival (38, 51). Our results are consistent with previous studies in which infection rates by *Francisella*-LE reached 97% in *H. marginatum* ticks collected in southern France (19) and 90% of *H. marginatum* pools collected in Corsica (4). The absence of influence of spatial groups on the rate of infection is not surprising, as it is a primary symbiont necessary for the tick's physiology. In contrast, the sex of ticks had a slight but significant influence on the presence of *Francisella*-LE with higher infection rates in females than in males. It is interesting to note that this observation has already been reported in *Dermacentor* ticks (20, 22). Although *Francisella*-LE quantification was not estimated in our study, this difference may also be explained by the fact that *Francisella*-LE loads might be lower in males and therefore make the detection more difficult, as already demonstrated in *Dermacentor variabilis* ticks (52). Neither the engorgement status of females nor the host of the tick had any influence on the presence of *Francisella*-LE.

*A. phagocytophilum* was detected in only 1.6% of the collected ticks. *A. phagocytophilum* is the etiological agent of human granulocytic anaplasmosis, tick-borne fever affecting ruminants, and equine granulocytic anaplasmosis (53–55). This bacterium is maintained in wild mammals (roe-deer, white-tail deer, white-footed mice), domestic animals (cattle, sheep), and birds in an enzootic cycle which is consistent with our results since 7/8 of *A. phagocytophilum* infected ticks were collected from cattle. Since *A. phagocytophilum* is considered an emerging pathogen of horses (56), it is not surprising to find a tick collected from a horse infected by this bacterium. Finally, most of the infected ticks were females (6/8) that were either semi-engorged or fully engorged although no significant influence of the tick sex or the engorgement status was observed. This result could indicate that ticks become infected after a blood meal,

but the role of *H. marginatum* as a vector of this pathogen is questioned and further studies need to address its vector competence (19). Interestingly, another *Anaplasma* species, *A. marginale,* was detected in ticks with an infection rate of 0.8%. *A. marginale* is responsible for fever, anemia, weight loss, and abortions in cattle. The main reported vectors of this bacteria are *Rhipicephalus* spp. and *Dermacentor* spp. in tropical and subtropical regions that can be found in Europe. Statistics were not applicable due to the low number of *A. marginale* infected ticks ($n = 4$) but descriptive data showed that all infected ticks were engorged females (semi or fully-fed). All the ticks were collected in two sites of Aude/Pyrénées-Orientales, on cattle. The most likely hypothesis is that the four female-fed ticks infected with *A. marginale* became infected by blood-feeding on the same infected animal bovine host.

Finally, the equine piroplasmosis agent *T. equi* parasite was detected in our study. We first estimated the *T. equi* infection rate at 1.8% with the BioMark assay. This approach made us aware of the probable underestimation of the infection rate as we used primers that did not allow the detection of all genotypes potentially circulating in France. We re-evaluated *T. equi* infection rates at 9.2% using another gene (18S rRNA) that can target several genotypes of *T. equi* by qPCR. Interestingly, this infection rate was very low compared to a study that showed that 43% of *H. marginatum* collected from horses in Camargue, next to the Occitanie region, were infected with *T. equi* (57). This large difference may probably be explained by the design of the tick collection that aimed to target stables where cases of piroplasmosis were frequently reported. Most *T. equi*-infected ticks were collected on horses (33/47). As horses are known to be reservoirs of *T. equi* and a high circulation rate of piroplasmosis has been described in France, it is not surprising that the majority of infected ticks were collected on horses (58). It should be noted that 14/47 of the infected ticks were collected from cattle, which is more surprising since cattle are not known to be susceptible to infection by *T. equi* but rather to *Theileria annulata* and *Theileria parva*. Such a result could suggest maternal transmission of this parasite in ticks but this has not yet been demonstrated and should be tested experimentally in *H. marginatum*. It could also suggest that the tick could have become infected during its immature stages on a susceptible host. It is currently unknown if the hosts of immature stages (birds and lagomorphs) can carry and be infected with *T. equi*. While *T. equi* infection rates were not influenced by the geographic cluster, significant differences were observed on *T. equi* loads, with a lower number of genomes.$\mu L^{-1}$ in Hérault/Gard than in Aude/Pyrénées-Orientales. This result has to be nuanced regarding the fact that a tick collected on cattle in a site of Aude/Pyrénées-Orientales presented very high *T. equi* loads compared to other infected ticks. Finally, although *T. equi* infection rates and loads were not influenced by the sex of the tick, its infection rate was significantly higher in fed females compared to unfed ones, suggesting that *H. marginatum* ticks tend to be infected by *T. equi* via the blood meal on a parasitemic horse rather than being responsible for the transmission of *T. equi* to an uninfected horse.

## Conclusion

This study characterized the spatial distribution of *H. marginatum*-borne pathogens in the region Occitanie where the tick has recently become established. At the scale of the whole region, we reported that (i) 11.8% of *H. marginatum* ticks were infected with at least one species of *Theileria* or *Anaplasma*; (ii) the infection rate of *R. aeschlimannii* was very high (87.3%) while few rickettsiosis cases are reported, which questions the vector capacity of *H. marginatum* for this bacterium, (iii) *R. aeschlimannii* is hypothesized to be a secondary symbiont of *H. marginatum* due to its high infection rates and its maternal transmission; (iii) infection rates of all detected pathogens were quite variable from one site of collection to another, demonstrating the importance of the sampling effort for pathogen surveillance. Beyond the spatial scale, pathogens dynamic can also fluctuate depending on the temporal scale, even at a monthly scale (14, 27). It is necessary to highlight temporal patterns that might affect pathogens and symbionts

in *H. marginatum*. To do this annual and monthly monitoring of ticks collected in a given location would help us identify temporal dynamics.

## MATERIALS AND METHODS

### Study area and tick collection

A large-scale tick sampling was conducted during 2 weeks in May 2022. From the Stachurski and Vial (59) study, we identified and visited 42 sites (riding schools and farms) from six French departments of the Occitanie region (Fig. 1). Ticks were collected by direct removal on the horses (*n* = 384) and cattle (*n* = 90). Ticks were morphologically identified using a binocular loupe, sorted by sex (male/female) and engorgement status (fed, semi-engorged, and unfed) and then stored at −80℃ until further use. After the identification, a total of 510 *H. marginatum* ticks were analyzed. They came from four departments of the Occitanie region: Hérault, Gard, Aude, and Pyrénées-Orientales whereas it was not detected in the Tarn and Haute-Garonne departments (Fig. 1). We chose a maximum of 34 ticks per site, with an average of 18.8 ticks per site. When the number of ticks was lower than 34, all ticks were analyzed. When feasible, an equal number of males and females was selected for each site.

### DNA and RNA extraction

Tick crushing and nucleic acid extraction were done in a BSL3 facility. Ticks were washed in hypochlorite solution 1% that was diluted from a bottle with 2.6% of active chlorine (Orapi Group, France) for 30 s then rinsed three times for 1 min in milli Q water to limit eliminate the environmental microbes present on the tick cuticle (60). Ticks were then cut using a scalpel blade and crushed individually in the homogenizer Precellys24 Dual (Bertin, France) at 5,500 rpm for 40 s, using three steel beads (2.8 mm, OZYME, France) in 400 µL of Dulbecco's Modified Eagle Medium (DMEM, Eurobio Scientific France) with 10% fetal calf serum. Total DNA and RNA were extracted using the NucleoMag VET extraction kit (Macherey-Nagel, Germany) as described by the manufacturer's instructions with the IDEAL 96 extraction robot (Innovative Diagnostics, France). Nucleic acids were eluted in 90 µL of elution buffer and stored at −20℃ for DNA and −80℃ for RNA until further analyses.

### Tick-borne pathogens detection in tick DNA and RNA

#### *Microfluidic PCR detection*

#### *Reverse transcription*

Sample were retrotranscribed in cDNA using 1 µL of RNA with 1 µL of Reverse Transcription Master Mix and 3 µL of RNase-free ultrapure water provided with the kit (Standard Biotools, USA) using a thermal cycler (Eppendorf, Germany) with the following cycles: 5 min at 25℃, 30 min at 42℃, and 5 min at 85℃ with a final hold at 10℃.

#### *Targeted tick-borne pathogens*

The tick-borne pathogens with a high probability of being carried by *H. marginatum* were targeted using 48 sets of primers (designs), according to data mining of the available literature (19, 61, 62) (Table 1). On the 48 sets of designs, 14, 5, and 27 targeted bacteria, parasites, and viruses, respectively. Two designs were used for positive controls. The 48 designs were pooled to reach a 200 nM final concentration for each primer. CCHFV detection on samples was performed apart from this study (13).

#### *Pre-amplification*

Each sample was pre-amplified using 1.25 µL of DNA mix (1:1 vol ratio of DNA and cDNA) with 1 µL of the PreAmp Master mix (Standard Biotools, USA), 1.5 µL of ultra-pure water

and 1.25 µL of the pooled designs. PCRs were performed using a thermal cycler with the following cycles: 2 min at 95°C, 14 cycles at 95°C for 15 s and 60°C for 4 min, and finally a 4 min hold at 4°C. A negative control was used for each plate with ultra-pure water. Amplicons were diluted 1:10 with ultra-pure water and stored at −20°C until further use.

### BioMark assay

The BioMark real-time PCR system (Standards Biotools, USA) was used for high-through-put microfluidic real-time PCR amplification using the 48.48 dynamic array (Standard Biotools, USA). The chips dispense 48 PCR mixes and 48 samples into individual wells, after which on-chip microfluidics assemble PCR reactions in individual chambers prior to thermal cycling resulting in 2,304 individual reactions. In one single experiment, 47 ticks and one negative control are being tested. For more details, please see references (61, 62).

### Validation of the results by PCR and sequencing

Conventional PCRs or qPCRs were then performed for tick-borne pathogens positive samples using different sets of primers than those used in the BioMark assay to confirm the presence of pathogenic DNA (Table S1). Amplicons were sequenced by Azenta Life Sciences (Germany) using Sanger-EZ sequencing and assembled using the Geneious software (Biomatters, New Zealand). An online BLAST (National Center for Biotechnology Information) was done to compare results with published sequences in GenBank sequence databases (4, 61–63).

## Detection and quantification of *T. equi* and *R. aeschlimannii* by duplex real-time fluorescence quantitative PCR

Tick samples were also screened for the detection and quantification of *T. equi* and *R. aeschlimannii* using a second detection method by qPCR with primers and probes targeting different genes to those used in the BioMark assay (Table 2) (57, 67). There are five known genotypes for *T. equi* (designated A–E) circulating in Europe, so we wanted to be sure that we could detect all of those five genotypes if present (68–71). The Takyon No ROX Probe 2× MasterMix Blue dTTP (Eurogentec, Belgium) was used with a final reaction volume of 20 µL containing 10 µL of Master Mix 2× (final concentration 1×), 5 µL of RNase-free water, 1 µL of each primer (0.5 µM), probes (0.25 µM), and 2 µL of DNA template. The reaction was carried out using a thermal cycler according to the following cycles: 3 min at 95°C, 45 cycles at 95°C for 10 s, 55°C for 30 s, and 72°C for 30 s. Positive controls for both *R. aeschlimannii* and *T. equi* were prepared using a recombinant plasmid from the TA cloning kit (Invitrogen, USA). A 10-fold serial dilution of the plasmid (from an initial concentration of $0.5 \times 10^8$ copy number/µL) was used to generate standard positive plasmids from $2.5 \times 10^5$ copy number/µL to $2.5 \times 10^{-1}$ copy number/µL. Samples were detected in duplicates and quantified using the standard plasmids. For *R. aeschlimannii*, we considered negative samples whose Cq number was higher than Cq 37. This detection limit was established regarding the last dilution of the standard curve that could be detected by qPCR. For *T. equi*, most samples were close to or below the detection limit established with the *T. equi* standard curve. Because the protozoan *T. equi* may be circulating at low levels, we included all positive samples.

## Statistical analyses

Statistical analyses were performed with R software 4.2.0. Generalized linear mixed effect models [glmer, package lme4 (73)] were used to evaluate the effect of variables (geographic cluster, tick sex, engorgement status, site of collection, and host of the tick) on the presence/absence (binomial distribution) of *R. aeschlimannii*, *A. phagocytophilum*, *Francisella*-LE (BioMark data), and *T. equi* (qPCR data). In addition, the influence of the geographic cluster, tick sex, engorgement status, and the site of the collection were also

**TABLE 1** Tick-borne pathogens targeted and the specific genes (designs) using the high-throughput microfluidic system (Biomark assay)

| Genus/family | Species | Targeted gene | Name | Sequence |
|---|---|---|---|---|
| Anaplasma | A. marginale | msp1b | An_ma_msp1_F | CAGGCTTCAAGCGTACAGTG |
| | | | An_ma_msp1_R | GATATCTGTGCCTGGCCTTC |
| | | | An_ma_msp1_P | ATGAAAGCCTGGAGATGTTAGACCGAG |
| | A. phagocytophilum | msp2 | An_ph_msp2_F | GCTATGGAAGGCAGTGTTGG |
| | | | An_ph_msp2_R | GTCTTGAAGCGCTCGTAACC |
| | | | An_ph_msp2_P | AATCTCAAGTCAACCCTGGCACCAC |
| | Anaplasma spp. | 16S rRNA | Ana_spp_16S_F | CTTAGGGTGTAAAACTCTTCAG |
| | | | Ana_spp_16S_R | CTTTAACTTACCAAACCGCCTAC |
| | | | Ana_spp_16S_P | ATGCCCTTTACGCCCAATAATTCCGAACA |
| Bartonella | B. henselae | Pap31 | Bar_he_pap31_F | CCGCTGATCGCATTATGCCT |
| | | | Bar_he_pap31_R | AGCGATTTCTGCATCATCTGCT |
| | | | Bar_he_pap31_P | ATGTTGCTGGTGGTGTTTCCTATGCAC |
| | Bartonella spp. | ssrA | Bart_spp_ssrA_F | CGTTATCGGGCTAAATGAGTAG |
| | | | Bart_spp_ssrA_R | ACCCCGCTTAAACCTGCGA |
| | | | Bart_spp_ssrA_P | TTGCAAATGACAACTATGCGGAAGCACGTC |
| Borrelia | B. miyamotoi | glpQ | B_miya_glpQ_F | CACGACCCAGAAATTGACACA |
| | | | B_miya_glpQ_R | GTGTGAAGTCAGTGGCGTAAT |
| | | | B_miya_glpQ_P | TCGTCCGTTTTCTCTAGCTCGATTGGG |
| | Borrelia spp. | 23S rRNA | Bo_bu_sl_23S_F | GAGTCTTAAAAGGGCGATTTAGT |
| | | | Bo_bu_sl_23S_R | CTTCAGCCTGGCCATAAATAG |
| | | | Bo_bu_sl_23S_P | TAGATGTGGTAGACCCGAAGCCGAGT |
| Coxiella | C. burnetii | idc | Co_bu_icd_F | AGGCCCGTCCGTTATTTTACG |
| | | | Co_bu_icd_R | CGGAAAATCACCATATCCACCTT |
| | | | Co_bu_icd_P | TTCAGGCGTTTTGACCGGGCTTGGC |
| | Coxiella-like symbionts | IS1111 | Co_bu_IS111_F | TGGAGGAGCGAACCATTGGT |
| | | | Co_bu_IS111_R | CATACGGTTTGACGTGCTGC |
| | | | Co_bu_IS111_P | ATCGGACGTTTATGGGGATGGGTATCC |
| Ehrlichia | Ehrlichia spp. | 16S rRNA | Neo_mik_16S_F | GCAACGCGAAAAACCTTACCA |
| | | | Neo_mik_16S_R | AGCCATGCAGCCACCTGTGT |
| | | | Neo_mik_16S_P | AAGGTCCAGCCAAACTGACTCTTCCG |
| Francisella | F. tularensis | tul4 | Fr_tu_tul4_F | ACCCACAAGGAAGTGTAAGATTA |
| | | | Fr_tu_tul4_R | GTAATTGGGAAGCTTGTATCATG |
| | | | Fr_tu_tul4_P | AATGGCAGGCTCCAGAAGGTTCTAAGT |
| | Francisella-like symbionts | fopA | Fr_tu_fopA_F | GGCAAATCTAGCAGGTCAAGC |
| | | | Fr_tu_fopA_R | CAACACTTGCTTGAACATTTCTAG |
| | | | Fr_tu_fopA_P | AACAGGTGCTTGGGATGTGTGGGTGGTG |
| Rickettsia | R. aeschlimannii | 23 s-5S ITS | Rick_aesch_ITS_F | CTCACAAAGTTATCAGGTTAAATAG |
| | | | Rick_aesch_ITS_R | CTTAACTTTTACTACGATACTTAGCA |

**TABLE 1** Tick-borne pathogens targeted and the specific genes (designs) using the high-throughput microfluidic system (Biomark assay) (Continued)

| Genus/family | Species | Targeted gene | Name | Sequence |
|---|---|---|---|---|
| | Rickettsia spp. | gltA | Rick_aesch_ITS_P | TAATTTTGCTGGATATCGTGGCGGGG |
| | | | Rick_spp_gltA_F | GTCGCAAATGTCACGGTACTT |
| | | | Rick_spp_gltA_R | TCTTCGTGCATTTCTTTCCATTG |
| | | | Rick_spp_gltA_P | TGCAATAGCAAGAACCGTAGGCTGGATG |
| Babesia | B. caballi | Rap1/48 kDa mero antigen | Ba_cab_rap1_F | GTTGTTCGGCTGGGGCATC |
| | | | Ba_cab_rap1_R | CAGGCGACTGACGCTGTGT |
| | | | Ba_cab_rap1_P | TCTGTCCCGATGTCAAGGGGCAGGT |
| | B. canis | 18S rRNA | Ba_ca_RNA18S_F | TGGCCGTTCTTAGTTGGTGG |
| | | | Ba_ca_RNA18S_R | AGAAGCAACCGGAAACTCAAATA |
| | | | Ba_ca_RNA18S_P | ACCGGCACTAGTTAGCAGGTTAAGGTC |
| Theileria | T. equi | ema1 | Th_eq_ema1_F | GGCTCCGGCAAGAAGCACA |
| | | | Th_eq_ema1_R | CTTGCCATCGACGACCTTGA |
| | | | Th_eq_ema1_P | CTTCAAGGCTCCAGGCAAGCGCGT |
| | T. annulata | 18S rRNA | Th_an_18S_F | GCGGTAATTCAGCTCCAATA |
| | | | Th_an_18S_R | AAACTCCGTCCGAAAAAAGCC |
| | | | Th_an_18S_P | ACATGCACAGACCCCAGAGGGACAC |
| | Theileria spp. | 18S rRNA | Thei_spp_18S_F | GTCAGTTTTACGACTCCTTCAG |
| | | | Thei_spp_18S_R | CCAAAGAATCAAGAAAGAGCTATC |
| | | | Thei_spp_18S_P | AATCTGTCAATCCTTCCTTGTCTGGACC |
| Bunyaviridae | Nairobi sheep disease virus | segment M1 | Nairobi_M1_F | CACCAGTGCATTTACACACAAC |
| | | | Nairobi_M1_R | GTTCCTTTCATCAGTCAGGTTG |
| | | | Nairobi_M1_P | ATGCAGATGGGCAAGTGAGAGTACCCA |
| | | segment M2 | Nairobi_M2_F | ACAGTGGTACAAGGAGAACCA |
| | | | Nairobi_M2_R | AGCCTTCTTTTGCCTGTCACA |
| | | | Nairobi_M2_P | TTGGTGCGTATACAATGAGCTTCAACTTGACT |
| | | segment G1 | NSDV G1 F | TCTAAGTGCTAGCCCTGATGT |
| | | | NSDV G1 R | GCCAACTGAGTGTTCTTCTC |
| | | | NSDV G1 P | TTCTACAGGCCGTCCGTCAAGGAAGA |
| | | segment G1 bis | NSDV G1bis F | ACTAAGTGCAAGCTCAGAAGC |
| | | | NSDV G1bis R | ACCCACAGAATGTTCATCCTC |
| | | | NSDV G1bis P | TCCTACTGTGTGTCCTTCAGGGGTTG |
| Flaviviridae | West Nile virus | Polyprotein gene | WN_F | AAGTTGAGTAGACGGTGCTGC |
| | | | WN_R | AGACGGTTCTGAGGGCTTAC |
| | | | WN_P | CGACTCAACCCAGGAGGACTGG |
| | | segment 1A | WN_1A_F | GTTGGCTCTTGGCGTTCT |
| | | | WN_1A_R | GCAATTCCGGTCTTTCCTCC |
| | | | WN_1A_P | TCAGGTTCACAGCAATTGCTCCGACC |
| | | segment 1B | WN_1B_F | GAAGTTAGCAGTCTACGTTAGG |

**TABLE 1** Tick-borne pathogens targeted and the specific genes (designs) using the high-throughput microfluidic system (Biomark assay) (*Continued*)

| Genus/family | Species | Targeted gene | Name | Sequence |
|---|---|---|---|---|
| | | | WN_1B_R | GCATATCCAGGGTTTCTCAAG |
| | | | WN_1B_P | TATGGAAGATGCACCAAGACACGACACTC |
| | | segment 1C | WN_1C_F | TCATGGTTGCGACGTTCGTG |
| | | | WN_1C_R | AAGTGTTGGTAAACGTGATGGC |
| | | | WN_1C_P | AAGGCTAGGTGGCGAACCAGGAGAA |
| | | segment 2.1 | WN_2.1_F | GAGCTGTTTCTTAGCACGAAG |
| | | | WN_2.1_R | CAGACTCAGCATAGCCCTCT |
| | | | WN_2.1_P | ATCTGATGTCTAAGAAACCAGGAGGGC |
| | | segment 2.2 | WN_2.2_F | CATGGAGAAAGTACACTGGCTA |
| | | | WN_2.2_R | GCAGTAGGATAGCGAACACG |
| | | | WN_2.2_P | ATAAGAAAGGAGCTTGGCTGGACAGCAC |
| | | segment 3 | WN_3_F | ATTTGAAGAACCACATGCCACG |
| | | | WN_3_R | TGCGCATACTCCATAGGTCG |
| | | | WN_3_P | AAGCAATCGGTGGTCGCCTTAGGTTCT |
| | | segment 4 | WN_4_F | GATTGTGAACCCAGGTCAGG |
| | | | WN_4_R | TGTTCCTCCAGTTCGTGTTTC |
| | | | WN_4_P | CGTTGATGTGGACGCCTTCTACGTGAT |
| *Nairoviridae* | Alkhurma virus (Kyasanur) | C gene | Alkhurma_C_F | ATTGCCAAATGGACTGGTGTTG |
| | | | Alkhurma_C_R | ACCTTCTTCCCTCTTCTTCT |
| | | | Alkhurma_C_P | ATGCGCATGATGGGAGTGTTGTGGCA |
| | Hazara virus | segment L2 | Hazara_L2_F | GCAGAGGCTATGTGCAGGTA |
| | | | Hazara_L2_R | TAAACTCTGTGCAACACCGGA |
| | | | Hazara_L2_P | CACATCCACGCCATACCAAGAAATACCC |
| | Dugbe virus | segment S | Dugbe_segS_F | GCACAAGGAGCACAAATAGAC |
| | | | Dugbe_segS_R | TTTTTGCCTCCTCTAGCACTC |
| | | | Dugbe_segS_P | TGGCCCATCTCAAAGAGGAATTGAGAC |
| *Orthomyxoviridae* | Thogoto virus | segment 6 | Thogoto_seg6_F | GGTCCTCAAGAAACGTCAGCA |
| | | | Thogoto_seg6_R | CATGTAAGTACCAAGACTCATCG |
| | | | Thogoto_seg6_P | AAAGTCGCCCTTCTCCGGGAAAGCAT |
| | | | Thogoto2_seg6_F | CTACAGTCAAGCAACCTCACTA |
| | | | Thogoto2_seg6_R | CCAGTAGACAGAGATTGGACA |
| | | | Thogoto2_seg6_P | CTACAGTGCAAGTTATGATGGTATACCGGTT |
| | Dhori virus | segment 2 | Dhori_seg2_F | CAAGCTCTGGTGTGCCTGT |
| | | | Dhori_seg2_R | CAGTTACTTCTGAGACAGCCT |
| | | | Dhori_seg2_P | AGGAGGGGAAGAGAAGTTGGCCAAG |
| *Phenuiviridae* | Bhanja virus | *segment L1* | Bhanja_L1_F | CTCTCAGGAAGAATCAAGGTGA |
| | | | Bhanja_L1_R | GAGAGGCCATCCAAATCTTCT |
| | | | Bhanja_L1_P | AGCCTTGACGCAGATGACCACACTTCA |

(*Continued on next page*)

**TABLE 1** Tick-borne pathogens targeted and the specific genes (designs) using the high-throughput microfluidic system (Biomark assay) (Continued)

| Genus/family | Species | Targeted gene | Name | Sequence |
|---|---|---|---|---|
| | | segment L2 | Bhanja_L2_F | CCATTATCAGACACCACAGGTA |
| | | | Bhanja_L2_R | ATGGAATCTCGAAGTCCTTGAG |
| | | | Bhanja_L2_P | TTGAGTCATGCCTAGAGGGCAACAACTG |
| | | segment L3 | Bhanja_L3_F | AGGTGTCTTGTGAATCTCAAGG |
| | | | Bhanja_L3_R | TTGACAGCTCAGTAGCAAGTTG |
| | | | Bhanja_L3_P | ACTTAGGGATTCTGTGTAACCACCTTCAAGA |
| | | segment L4 | Bhanja_L4_F | AGCATTGACTCTGGCTGAGG |
| | | | Bhanja_L4_R | CCTAAAGCTCTCATCTGCACTT |
| | | | Bhanja_L4_P | AGTCTGCCATCTTTGGTGACATAGTGTCTTT |
| | | Palma S | Bhanja_Palma_S_F | GCGCTGTTTGCCTATCAAGG |
| | | | Bhanja_Palma_S_R | TGCTTTTCAGGGCCTTCAATCT |
| | | | Bhanja_Palma_S_P | CTTTGACCCAACCAGGATGCTGAAGAAG |
| | | Forecariah S | Bhanja_Forecariah_S_F | ACTCATGCCTTCTATCTCTGG |
| | | | Bhanja_Forecariah_S_R | AAGGTCTGAAGGACAATGGCTT |
| | | | Bhanja_Forecariah_S_P | CAATTCTACTTCACCCAACTGATCAACCCAA |
| | | Razdan L | Bhanja_Razdan_L_F | GATTGGCAGCAGCATAGCCT |
| | | | Bhanja_Razdan_L_R | TTCTCCGTCGTTCATCACCG |
| | | | Bhanja_Razdan_L_P | TGGGAGCCACAAGAATAGAAGTCGGC |
| | | Kismayo L | Bhanja_Kismayo_L_F | TTTCAGATCTTCGCCTAAGCC |
| | | | Bhanja_Kismayo_L_R | TTCTTTGCCCTTCCTTTCTGCT |
| | | | Bhanja_Kismayo_L_P | AGCAGACTGTTGATGAGCTAGTGTATAGGTT |
| | | Forecariah L | Bhanja_Forecariah_L_F | ATTTGCATGTGCTGTCATGGCT |
| | | | Bhanja_Forecariah_L_R | TGTCAAGTATCTCCTTCTCTCC |
| | | | Bhanja_Forecariah_L_P | CAGATTCTTGTGGCCTTCCCTGGTCTT |
| Tick species | | 16S rRNA | 16S rRNA F | AAATACTCTAGGGATAACAGCGT |
| | | | 16S rRNA R | TCTTCATCAAACAAGTATCCTAATC |
| | | | 16S rRNA P | CAACATCGAGGTCGCAAACCATTTTGTCTA |
| | Escherichia coli | eae | eae F | CATTGATCAGGATTTTCTGGTGATA |
| | | | eae R | CTCATGCGGAAATAGCCGTTA |
| | | | eae P | ATAGTCTCGCCAGTATTCGCCACCAATACC |

<sup>a</sup>See references (19, 61, 63–66) for more details.

**TABLE 2** *R. aeschlimannii* and *T. equi* primers and probes sequences used for the detection and quantification method by duplex real-time quantitative PCR

| Pathogen | Targeted gene | Primer name | Sequence (5′–3′) | Reference |
|---|---|---|---|---|
| *R. aeschlimannii* | *ompB* | aes280F | CATCACTAATGCTGCTAATAACG | (72) |
| | (outer membrane protein B) | aes280R | CAGCAGCTTGTGCATTAGTAATA | |
| | | aes307P | YY-CTCCTTTGACCTTAGGGTTGATGCCG-BHQ | |
| *T. equi* | 18S rRNA | Be18SF | GCGGTGTTTCGGTGATTCATA | (57) |
| | | Be18SR | TGATAGGTCAGAAACTTGAATGATACATC | |
| | | Be18SP | FAM-AAATTAGCGAATCGCATGGCTT-BHQ | |

assessed on the loads (qPCR data) of the positive samples for *R. aeschlimannii* and *T. equi* using a glmer with a gamma distribution. The presence and loads of each pathogen were analyzed according to three types of models as follows: A first was used on the whole data set to assess the influence of the geographic cluster (Aude/Pyrénées-Orientales and Hérault/Gard), the tick sex (male and female), and the site of the collection as a random effect. A second model was used to evaluate the influence of the engorgement status, with a subset including females only, since the engorgement status of males could not be evaluated. Maximal models included the engorgement status, the geographic cluster the site of the collection as a random effect. Finally, a third model was used to assess the influence of the host on a subset corresponding to ticks located in the Aude/Pyrénées-Orientales cluster, since ticks were collected on both horses and cattle, while they were only collected from horses in Hérault/Gard. Maximal models included the host, the tick sex, and the site of the collection as a random effect. Minimal models and significance of variables were assessed using the "analysis of variance" procedure within the package "car" which performs a type III hypothesis (74). Post hoc tests were conducted using the function "emmeans" (Tukey HSD test). The *P*-value associated with the random effect "site" was assessed by log-likelihood test in the first model. Because very few ticks were positive for *A. marginale*, only descriptive analyses were presented in Table S2. We decided to include *Francisella*-LE in the analyses as a control since this bacterium is identified as an *H. marginatum* primary symbiont.

## Maternal transmission of *R. aeschlimannii*

To assess the potential maternal transmission of the bacteria *R. aeschlimannii*, five fed females collected on the field were individually placed in 50 mL Falcon tubes in an incubator at a temperature of 27°C for several weeks in the BSL3 facility. About 100 eggs belonging to each of the five females were isolated and frozen at −80°C for subsequent analyses while the rest of the eggs were left in the incubator. Only a few eggs (from one adult female) hatched into larvae (*n* = 12). The eggs and larvae were placed separately in a 0.2 mL microtube with 100 µL of DMEM and crushed against the bottom of the tube using a sterile needle. DNA was extracted using 100 µL of homogenate with the Genomic DNA tissue kit (Macherey-Nagel, Germany). Detection and quantification of *R. aeschlimannii* was performed by targeting the *ompB* gene by qPCR (Table 2).

## ACKNOWLEDGMENTS

The authors are grateful to Maud Marsot, Facundo Munoz, Maxime Prat, and Haoues Alout for their helpful advice on statistics. We are grateful to Mickaël Mège and Claire Bonsergent for technical assistance and Frédéric Stachurski for the relevant discussions.

The work was funded by the Holistique project (défi clé RIVOC Occitanie region, University of Montpellier): "*Hyalomma marginatum* in Occitanie region: analysis of biological invasion and associated risks".

Conceptualization: T.P., S.M., C.J.K. Funding acquisition: T.P. Methodology: C.J.K., C.G., D.B., C.R., S.M. Formal analysis: C.J.K. Field and resources: C.J.K., C.B., H.J.P., K.H., I.K., L.M., F.S., L.V., T.P. Writing-original draft: C.J.K., T.P., S.M. All authors read and approved the final manuscript.

## AUTHOR AFFILIATIONS

¹ASTRE, Univ Montpellier, CIRAD, INRAE, Montpellier, France
²ANSES, INRAE, Ecole Nationale Vétérinaire d'Alfort, UMR BIPAR, Laboratoire de Santé Animale, Maisons-Alfort, France
³CIRAD, UMR ASTRE, Montpellier, France
⁴French Establishment for Fighting Zoonoses (ELIZ), Malzéville, France
⁵BIOEPAR, INRAE, Oniris, Nantes, France

## AUTHOR ORCIDs

Charlotte Joly-Kukla http://orcid.org/0009-0009-2905-8234
Sara Moutailler http://orcid.org/0000-0003-3010-6968

## AUTHOR CONTRIBUTIONS

Charlotte Joly-Kukla, Conceptualization, Data curation, Formal analysis, Investigation, Methodology, Resources, Validation, Visualization, Writing – original draft, Writing – review and editing | Célia Bernard, Resources, Writing – review and editing | David Bru, Methodology | Clémence Galon, Methodology, Writing – review and editing | Carla Giupponi, Resources, Writing – review and editing | Karine Huber, Resources, Writing – review and editing | Hélène Jourdan-Pineau, Methodology, Resources, Writing – review and editing | Laurence Malandrin, Resources, Writing – review and editing | Ignace Rakotoarivony, Resources | Camille Riggi, Methodology | Laurence Vial, Resources, Writing – review and editing | Sara Moutailler, Conceptualization, Investigation, Methodology, Resources, Supervision, Validation, Writing – original draft, Writing – review and editing | Thomas Pollet, Conceptualization, Funding acquisition, Investigation, Methodology, Project administration, Resources, Supervision, Validation, Writing – original draft, Writing – review and editing

## ADDITIONAL FILES

The following material is available online.

### Supplemental Material

**Supplemental material (Spectrum01256-24-S0001.pdf).** Tables S1 to S3.

### Open Peer Review

**PEER REVIEW HISTORY (review-history.pdf).** An accounting of the reviewer comments and feedback.

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
