## [Reviewer comments · Microbiology Spectrum]

Microbiology Spectrum

Spatial patterns of *Hyalomma marginatum*-borne pathogens in the Occitanie region (France), a focus on the intriguing dynamics of *Rickettsia aeschlimannii*

Charlotte Joly-Kukla, Célia Bernard, David Bru, Clemence Galon, Carla Giupponi, Karine Huber, H el ene Jourdan, Laurence Malandrin, Ignace Rakotoarivony, Camille Riggi, Laurence Vial, Sara Moutailler, and Thomas Pollet

Corresponding Author(s): Charlotte Joly-Kukla, Institut National de Recherche pour l'Agriculture l'Alimentation et l'Environnement Centre Occitanie-Montpellier

Review Timeline:

Submission Date:	May 27, 2024
Editorial Decision:	June 22, 2024
Revision Received:	June 27, 2024
Accepted:	June 28, 2024

Editor: Maristela Camargo

Reviewer(s): The reviewers have opted to remain anonymous.

Transaction Report:

DOI: <https://doi.org/10.1128/spectrum.01256-24>

Re: Spectrum01256-24 (Spatial patterns of *Hyalomma marginatum*-borne pathogens in the Occitanie region (France), a focus on the intriguing dynamics of *Rickettsia aeschlimannii*)

Dear Dr. Charlotte Joly-Kukla:

I am pleased to inform you that your manuscript has been editorially accepted for publication. However, there are a few additional questions in the submission form that need to be answered before the final decision. Once these are completed, please return your submission so that I can move your paper forward to acceptance.

Revision Guidelines

Sincerely,
Maristela Camargo
Editor
Microbiology Spectrum

Re: Spectrum01256-24R1 (Spatial patterns of *Hyalomma marginatum*-borne pathogens in the Occitanie region (France), a focus on the intriguing dynamics of *Rickettsia aeschlimannii*)

Dear Dr. Charlotte Joly-Kukla:

Your manuscript has been accepted, and I am forwarding it to the ASM production staff for publication. Your paper will first be checked to make sure all elements meet the technical requirements. ASM staff will contact you if anything needs to be revised before copyediting and production can begin. Otherwise, you will be notified when your proofs are ready to be viewed.

Sincerely,
Maristela Camargo
Editor
Microbiology Spectrum